# Extracellular Vesicle Biomarkers Reveal Inhibition of Neuroinflammation by Infliximab in Association with Antidepressant Response in Adults with Bipolar Depression

**DOI:** 10.3390/cells9040895

**Published:** 2020-04-06

**Authors:** Rodrigo B. Mansur, Francheska Delgado-Peraza, Mehala Subramaniapillai, Yena Lee, Michelle Iacobucci, Nelson Rodrigues, Joshua D. Rosenblat, Elisa Brietzke, Victoria E. Cosgrove, Nicole E. Kramer, Trisha Suppes, Charles L. Raison, Sahil Chawla, Carlos Nogueras-Ortiz, Roger S. McIntyre, Dimitrios Kapogiannis

**Affiliations:** 1Mood Disorders Psychopharmacology Unit, University Health Network, Toronto, ON M5T 2S8, Canada; m.subram@mail.utoronto.ca (M.S.); yenalee.lee@mail.utoronto.ca (Y.L.); mich.iacobucci@gmail.com (M.I.); nelson.rodrigues@crtce.com (N.R.); Joshua.Rosenblat@uhn.ca (J.D.R.); elisa.brietzke@queensu.ca (E.B.); Roger.McIntyre@uhn.ca (R.S.M.); 2Department of Psychiatry, University of Toronto, Toronto, ON M5T 2S8, Canada; 3Laboratory of Clinical Investigation, Intramural Research Program, National Institute on Aging, National Institutes of Health (NIA/NIH), Baltimore, MD 20892, USA; francheska.delgado-peraza@nih.gov (F.D.-P.); sahil.chawla@nih.gov (S.C.); carlos.nogueras-ortiz@nih.gov (C.N.-O.); kapogiannisd@mail.nih.gov (D.K.); 4Institute of Medical Science, University of Toronto, Toronto, ON M5S 1A8, Canada; 5Kingston General Hospital, Providence Care Hospital, Department of Psychiatry, Queen’s University School of Medicine, Kingston, ON K7L 4X3, Canada; 6Department of Psychiatry & Behavioral Sciences, Stanford University, School of Medicine, Palo Alto, CA 94304, USA; veileen@stanford.edu (V.E.C.); nklange@stanford.edu (N.E.K.); tsuppes@stanford.edu (T.S.); 7School of Human Ecology, University of Wisconsin-Madison, Madison, WI 53706, USA; raison@wisc.edu; 8Department of Psychiatry, School of Medicine and Public Health, University of Wisconsin-Madison, Madison, WI 30322, USA

**Keywords:** bipolar disorder, inflammation, cytokines, childhood trauma, depression

## Abstract

Accumulating evidence suggests that neuroinflammation is involved in bipolar disorder (BD) pathogenesis. The tumor necrosis factor-alpha (TNF-α) antagonist infliximab was recently reported to improve depressive symptoms in a subpopulation of individuals with BD and history of childhood maltreatment. To explore the mechanistic mediators of infliximab’s effects, we investigated its engagement with biomarkers of cellular response to inflammation derived from plasma extracellular vesicles enriched for neuronal origin (NEVs). We hypothesized that infliximab, compared to placebo, would decrease TNF-α receptors (TNFRs) and nuclear factor-kappa B (NF-κB) pathway signaling biomarkers, and that history of childhood abuse would moderate infliximab’s effects. We immunocaptured NEVs from plasma samples collected at baseline and at weeks 2, 6, and 12 (endpoint) from 55 participants of this clinical trial and measured NEV biomarkers using immunoassays. A subset of participants (*n* = 27) also underwent whole-brain magnetic resonance imaging at baseline and endpoint. Childhood physical abuse moderated treatment by time interactions for TNFR1 (χ^2^ = 9.275, *p* = 0.026), NF-κB (χ^2^ = 13.825, *p* = 0.003), and inhibitor of NF-κB (IκBα)α (χ^2^ = 7.990, *p* = 0.046), indicating that higher levels of physical abuse were associated with larger biomarker decreases over time. Moreover, the antidepressant response to infliximab was moderated by TNFR1 (χ^2^ = 7.997, *p* = 0.046). In infliximab-treated participants, reductions in TNFR1 levels were associated with improvement of depressive symptoms, an effect not detected in the placebo group. Conversely, reductions in TNFR1 levels were associated with increased global cortical thickness in infliximab- (r = −0.581, *p* = 0.029), but not placebo-treated, patients (r = 0.196, *p* = 0.501). In conclusion, we report that NEVs revealed that infliximab engaged the TNFR/NF-κB neuro-inflammatory pathway in individuals with BD, in a childhood trauma-dependent manner, which was associated with clinical response and brain structural changes.

## 1. Introduction

Replicated evidence indicates that inflammation is a relevant pathophysiological mechanism for a subset of individuals with bipolar disorder (BD). Multiple meta-analyses have documented alterations in peripheral markers of inflammation, such as acute phase proteins (e.g., C-reactive protein [CRP]) and inflammatory cytokines (e.g., tumor necrosis factor-α [TNF-α], interleukin-6 [IL6]) [1,2]. Direct evidence of neuroinflammation has been scarcer, but has supported the hypothesis of inflammatory dysregulation [3,4,5]. In addition, inflammatory mediators have been consistently associated with neurostructural abnormalities in individuals with BD [6,7,8,9,10]. As a result, anti-inflammatory agents have been recently tested as therapies for mood disorders, including bipolar depression [11,12,13]. Results have been mixed. For example, whereas anti-cytokine treatments have shown antidepressant activity in chronic inflammatory conditions [13], two recent clinical trials failed to demonstrate this effect in the intent-to-treat cohorts with treatment-resistant major depressive disorder and BD [14,15].

Nonetheless, both studies identified subpopulations, in secondary analyses, who were more likely to respond to the TNF-α antagonist infliximab, based on biochemical (e.g., elevated plasma CRP) or phenotypic criteria (e.g., exposure to early childhood adversity) [14,15]. It remains unclear, however, if these responders represent actual “inflammatory biotypes”, within or across mood disorders, which may be more responsive to anti-inflammatory interventions. Supporting this idea, a follow-up analysis of the study of Raison et al. [14], which identified elevated CRP at baseline as a predictor of favorable response to infliximab, documented that changes in the expression of genes related to innate immune signaling and nuclear factor-kappa B (NF-κB) in peripheral blood mononuclear cells (PBMCs) was associated with antidepressant response [16]. In our recently completed randomized, double-blind, placebo controlled, clinical trial evaluating infliximab for the treatment of bipolar depression, we identified early life trauma (i.e., childhood physical abuse) as a predictor of response [15]. Replicated evidence indicates that early childhood adversity is associated with elevated peripheral immuno-inflammatory activity in adulthood [17,18]. However, no studies so far have directly investigated neuroinflammatory processes in this population, nor potential mechanistic changes related to a therapeutic intervention.

A methodological challenge for the study of inflammation in brain disorders in humans has been the tenuous link between biomarkers and brain biochemistry and pathology. The majority of studies have used peripheral biomarkers to proxy brain processes, relying on the assumption that peripheral and brain immune activation are correlated (or that peripheral measures are reliable indicators of central processes), which has important limitations. A recent innovation is the use of blood extracellular vesicles (EVs) enriched for neuronal origin (NEVs) as a source of biomarkers that directly reflect neuronal-specific processes [19,20,21,22,23]. Extracellular vesicles (including exosomes) are membranous particles that are continuously released by virtually all cells, including neurons [24]. They can cross the blood–brain barrier and be preferentially isolated from peripheral blood, through immunocapture with antibodies against neuronal markers, most commonly L1 Cell Adhesion Molecule (L1CAM), a procedure that offers a multi-fold enrichment in neuronal cargo [19,20,21,22,23]. Extracellular vesicles contain cargo that reflects the content of the vesicle’s cellular origin, providing a novel avenue for probing intracellular signaling processes including neuroinflammatory pathways that are putatively implicated in neuropsychiatric disorders. This methodology has been used to directly assess neuronal inflammatory biomarkers in mild traumatic brain injury [25,26], as well as biomarkers from other pathways in disparate conditions, such as Alzheimer’s disease (AD) and schizophrenia [27,28]. In addition, NEVs have been used as a source of biomarkers that predict neuronal-specific molecular target engagement and response to experimental treatments in clinical trials (e.g., in Parkinson’s disease [21], AD [29], and cancer [30]), revealing concerted engagement of entire signaling pathways.

We therefore aimed to investigate the neuroinflammatory mediators of response to infliximab in individuals with bipolar depression. We focused herein on the TNF receptors (TNFR) 1 and 2 and the NF-κB pathway, given its role as effectors of TNF-α signaling and downstream transcriptional response, as well as previously published work on the molecular effects of TNF-α antagonists [16,31,32,33]. Our primary hypothesis was that NEVs of infliximab-treated compared to placebo-treated participants would display changes in levels of TNFR/NF-κB signaling effectors, in association with antidepressant response. Based on the results of this controlled clinical trial of infliximab in bipolar depression [15], it is also an a priori hypothesis that a history of childhood abuse would moderate infliximab’s effects on TNFR/NF-κB pathway biomarkers. A secondary, exploratory hypothesis was that changes in TNFR/NF-κB pathway biomarkers would be associated with changes in global cortical thickness.

## 2. Materials and Methods

### 2.1. Study Design

The clinical study was a 12-week randomized, double-blind, placebo-controlled, parallel-group, fixed-dose trial evaluating the efficacy, safety, and tolerability of adjunctive infliximab for the treatment of individuals with bipolar I/II depression meeting inflammatory criteria [15]. The study was approved by the Institutional Ethics Board at the University Health Network, Toronto, ON, Canada, as well as Stanford University, Palo Alto, California. The study is registered with ClinicalTrials.gov (Identifier: NCT02363738 5 January 2015).

### 2.2. Participants

Eligible participants were male and female outpatients between the ages of 18 and 65 who met Diagnostic and Statistical Manual of Mental Disorders, fifth edition (DSM-5) criteria for a current major depressive episode as part of bipolar I/II disorder. All subjects provided written informed consent after receiving a complete description of the study. The diagnosis of BD was confirmed using the Mini-International Neuropsychiatric Interview (M.I.N.I.) 5.0.0 for the DSM-IV-TR and interview responses were compared to DSM-5 diagnostic criteria to confirm BD I/II diagnosis. A total score of 22 or higher on the Montgomery-Asberg Depression Rating Scale (MADRS) and a total score of less than 12 on the Young Mania Rating Scale (YMRS) was required for inclusion in the study. Participants were not excluded based on their past or current psychiatric medication history. The study sample was enriched for an inflammatory phenotype, according to criteria described in McIntyre et al. [15]. Briefly, participants were required to exhibit pre-treatment biochemical (i.e., peripheral CRP level ≥5 mg/L) and/or phenotypic (e.g., comorbid inflammatory bowel disorder, rheumatologic disorder or metabolic syndrome) evidence of inflammatory activation.

### 2.3. Procedures

Participants were enrolled at the Mood Disorders Psychopharmacology Unit (MDPU) in Toronto and Stanford University in Palo Alto between October 2015 and April 2018. Only individuals enrolled at the MDPU site had blood samples taken and composed the sample analyzed herein. All participants meeting eligibility criteria at screening were randomized to either receive intravenous infliximab (5 mg/kg) or placebo (saline solution matched to infliximab in color and consistency) administered adjunctively to their existing guideline-congruent pharmacologic regimen for BD. Eligible participants were required to maintain their medication regimen during the 4 weeks before study enrollment and throughout the course of the 12-week study.

The infliximab dose and infusion schedule selected for this study were adopted from the protocol of a previously published interventional clinical trial with infliximab [14]. Participants received an infusion of infliximab (5 mg/kg) or placebo, administered over a period of 120 min by a rheumatology infusion registered nurse, at weeks 0 (baseline), 2, and 6. All participants completed the baseline infusion within one month of completing the screening assessment. Infliximab and placebo were prepared and dispensed in a concealed 250 mL infusion bag matched in color and consistency by hospital pharmacists who did not have contact with any of the participants. The randomization schedule was computer generated in blocks of 6 by a research team member who had no contact with the participants.

### 2.4. Clinical Measures

Participants, outcome assessors, principal investigators, and infusion nurses were all blinded to treatment randomization. Self-reported childhood maltreatment (i.e., Childhood Trauma Questionnaire [CTQ]) was measured at baseline. The CTQ is a 28-item test that measures 5 types of maltreatment—emotional, physical, and sexual abuse, and emotional and physical neglect. Based on previous findings published by McIntyre et al. [15], only the CTQ total score and the subdomain of physical abuse (PA) were used, both as continuous variables. Depressive symptoms severity was measured using the MADRS at every study visit.

### 2.5. Neuroimaging

High resolution 3D T1-weighted images were acquired from 14 participants from the placebo cohort and 13 participants from the infliximab group. Patients were scanned at two time points in the study, first at week 0 (baseline) prior to receiving any infusions and then again at week 12. Images were obtained on a General Electric Signa HDxt 1.5-Tesla scanner at Toronto General Hospital in Toronto, Ontario, Canada. The scanning parameters for the 3D T1-weighted fast spoiled gradient echo sequence were as follows: slice thickness = 1 mm, repetition time = 10.74 ms, echo time = 4.20 msec, inversion time = 450 msec, matrix size = 256 mm × 256 mm, field of view = 220 mm, flip angle = 15°, voxel size = 0.86 × 0.85 × 1 mm^3^, and scan duration = 15 min 3 s. There was a total of 146 slices produced in the axial plane.

Reconstruction of patient cortical tissue was preformed using the Freesurfer (v6.0) imaging analysis suite, which is freely available online (https://surfer.nmr.mgh.harvard.edu/). The technical details of the procedures are described in prior publications [34,35]. Given that patients were scanned at two separate time points, images were automatically processed using the longitudinal stream of Freesurfer in order to extract reliable thickness estimates [36]. First, images were cross-sectionally processed for each timepoint using the default pipeline. Briefly, the preprocessing involves correcting for motion and magnetic inhomogeneities, removing non-brain tissue, registering the images to Talairach space and segmenting the white and gray matter tissues. The boundary between white matter and gray matter (i.e., the white surface) is first calculated and then inflated to determine the boundary between gray matter and cerebral spinal fluid (i.e., the pial surface). Once completed, all time points for a single patient are used to create an unbiased within-subject template for each patient, which is then run again through the Freesurfer pipeline described above. In the final step, the processed template is used to resample the individual time points, which further reduces variability across time. The global cortical thickness was obtained from the ‘Mean Thickness’ variable outputted by Freesurfer v6.0. This variable is a calculation, in millimeters, of the total thickness below the pial surface subtracted by the total thickness below the white surface. Therefore, it measures the cortical thickness of only grey matter, found within the pial and white surfaces, within the left and right hemispheres [37]. Moreover, it excludes any subcortical volumes and the cerebellum. Global cortical thickness values were then extracted from the output using the ‘aparcstats2table’ command and were analyzed in IBM SPSS Statistics for Windows, version 23 (IBM Corp., Armonk, NY, USA).

### 2.6. Isolation of Extracellular Vesicles Enriched for Neuronal Origin

All available plasma samples from the clinical study were used and all experimental procedures were conducted blindly. Venous blood draws were conducted between 7:00 a.m.–10:00 a.m. after a 12 h fasting at baseline, and weeks 2, 6, and 12. Blood was collected in EDTA polypropylene tubes and within 1 h centrifuged at 3000 rpm for 15 min at 4 °C; supernatant plasma was divided into 0.5 mL aliquots and stored at −80 °C until analysis, in accordance with guidelines for pre-analytical factors for blood collection and storage for future biomarker analysis [38,39]. Plasma aliquots were processed blindly by a National Institute on Aging investigators (FDP, SC, CNO) following the methods published by Mustapic et al. [22]. NEVs were immunocaptured from plasma by targeting the neuronal trans-membrane marker L1 Cell Adhesion Molecule (L1CAM), which is expressed in EVs, following a methodology extensively characterized by our group [19,20,22] and others [40,41,42]. To reduce contamination with soluble plasma proteins, concentrate plasma EVs and, thereby, improve the yield of immunoprecipitation, plasma was defibrinated using Thrombin (System Biosciences, Inc., Mountainview, CA, USA) and total EVs were sedimented with Exoquick™ (System Biosciences, Inc., Mountainview, CA, USA) according to the manufacturer’s instructions. Total EVs were re-suspended in 0.5 mL of Ultra-pure distilled water with the manufacturer-recommended concentration of protease and phosphatase inhibitors. To immunocapture L1CAM+ NEVs, the suspension was incubated for 1 h at 4 °C with 4 µg of mouse anti-human CD171 (L1CAM) biotinylated antibody (clone 5G3) (Thermo Scientific, Inc., Waltham, MA, USA), followed by incubation with 25 µL of Pierce™ Streptavidin Plus UltraLink™ Resin (Thermo Scientific, Inc., Waltham, MA, USA) for 30 min at 4 °C. After centrifugation at 800× *g* for 10 min at 4 °C and removal of supernatant, NEVs were eluted with 200 µL of 0.1 M glycine. Then, beads were sedimented by centrifugation at 4500× *g* for 5 min at 4 °C, and the supernatants containing NEVs were transferred to clean tubes. pH was immediately neutralized with 1 M tris-HCl, and samples underwent 2 freeze thaw cycles with M-PER™ protein extraction reagent (Thermo Scientific, Inc., Waltham, MA, USA) supplemented with protease and phosphatase inhibitors to lyse the NEVs. The final suspensions containing NEV proteins were stored at −80 °C. Samples were thawed and vortexed twice prior to protein measurements. An extensive report on reproducibility and quality control measures for the NEV isolation methodology, detailed characterization for NEVs (by Nanoparticle Tracking Analysis, Electron Microscopy, and Western Blot quantification of canonical EV markers), and multifaceted evidence for neuronal cargo enrichment was recently published and is not repeated here as redundant [20].

### 2.7. NEV Protein Quantification

We quantified phosphorylated NF-κB (Ser436), FADD (Ser194), IKKα/β (Ser177/Ser181) and IκBα (Ser32), as well as total protein levels of TNFR1 and c-Myc using the MILLIPLEX^®^ MAP 6-Plex NF-κB Magnetic Bead Signaling kit (cat. no. 48-630MAG) (EMD Millipore Corporation, Billerica, MA, USA). Plates were read using Luminex^®^ 200™ System and the xPOTENT^®^ acquisition software (Luminex Corporation, Austin, TX, USA). In addition, we measured TNFR2 using a MESO SCALE DISCOVERY^®^ (MSD) electrochemiluminescence plate assay (K151BJC), read using a MESO QuickPlex SQ120 imager and the workbench Software 4.0 (Meso Scale Discovery, Rockville, MD, USA). Finally, we quantified Alix (or else human programmed cell death 6-interacting protein (PDCD6IP) (cat. no. CSB-EL017673HU) (Cusabio Biotech Co., LTD, Houston, TX, USA), an established EV marker enriched in exosomes [42,43], to assess differential NEV yield. Alix plates were read using the Synergy™ H1 microplate reader set to 450 nm and the Gen5™ microplate data collection software (BioTek Instruments, Winooski, VT, USA). The optimum dilution for each assay was determined using serial dilutions of test samples. For Alix, lysed NEVs were diluted 1:4 with the supplied sample diluent. No other assay required sample dilution. For TNFR2 and Alix assays, the concentration was determined using a standard curve separately for each plate using standards provided by the manufacturer and the four-parameter logistic regression curve-fit. For phospho-protein assays, a standard curve could not be constructed and thus we analyzed the electrochemiluminescence signal for MSD phospho-assays and the fluorescence signal for the Milliplex panel.

All assays were conducted in duplicate and the mean coefficients of variation (CV) across plates were 6.72% (TNFR2), 12.98% (NF-κB), 12.65% (FADD), 14.33% (IKKα/β), 14.51% (IκBα), 13.50% (TNFR1), 12.54% (c-Myc), and 4.29% (Alix). Duplicate NEV isolates from a healthy participant were included as internal control (IC) on every plate to assess between-plate variability. The CVs for the IC were: 10.57% (TNFR2), 8.54% (NF-κB), 11.38% (FADD), 14.49% (IKKα/β), 10.55% (IκBα), 10.37% (TNFR1), 10.62% (c-Myc), and 39.93% (Alix). Therefore, between-plates CVs for the IC for all analytes were under 20%, except for Alix, likely due to procuring kits from different lots. The IC was used to determine a correction factor (IC signal for a given plate divided by the average of IC signals in all plates), which was used to normalize raw signals from each plate.

The limit of detection (LOD), defined as mean of the blank plus 2.5 the standard deviation (SD) of the blank, was calculated from the electrochemiluminescence signal for MSD phospho-assays, fluorescence signal for the Milliplex panel, and colorimetric signal for the Alix ELISA. The LOD was: 3.03 ng/mL for TNFR2, 48.16 for NF-κB, 52.68 for FADD, 46.42 for IKKα/β, 34.04 for IκBα, 34.43 for TNFR1, 53.11 for c-Myc, and 168.71 ng/mL for Alix. The lowest limit of quantification (LLOQ) (defined as the concentration of the standard with: (1) signal above the LOD, (2) CV among duplicates <15%, and (3) recovery >80% and <120%), was calculated for each plate for TNFR2 and Alix assays, and the mean LLOQ was used as the global LLOQ. The LLOQ was 10.27 ng/mL for TNFR2, and 293.78 ng/mL for Alix.

Even though CVs of NEV samples from the same isolation run in duplicate were <15% for all biomarkers, to be even more conservative in our analysis, we identified and excluded the samples that had CVs of duplicates ≥15%, separately for each assay (17 samples for TNFR2, 35 samples for NF-κB, 32 samples for FADD, 32 samples for IKKα/β, 37 samples for IκBα, 34 samples for TNFR1, 30 samples for c-Myc, and 5 samples for Alix). Therefore, reported results are likely a conservative underestimation of true effect sizes, given the loss of power from excluding these values. Additional censoring criteria consisted of LLOQs for assays that had standard curve. For Alix, all samples were above the LLOQ and within the linear range of the standard curve For TNFR2, 2 samples were below the LLOQ and above the LOD, but were excluded from the analysis as they had a CV ≥15%. For MSD phospho-assays and MILLIPLEX^®^ 6-Plex, given that no standard curve was available, we excluded samples below the LOD: 1 for NF-κB, 10 for FADD, 1 for TNFR1, and 9 for c-Myc. TNFR2, IKKα/β, IκBα, and Alix did not had any sample below their respective LODs.

### 2.8. Statistical Analysis

To evaluate between-group differences in demographic baseline characteristics, non-parametric (i.e., Mann–Whitney U) and chi-square tests were used. A two-tailed alpha value of 0.05 was used to denote statistical significance. An intent-to-treat analysis (i.e., all participants who were randomized) was used to analyze changes over time of biomarkers’ levels. Biomarker values were natural log transformed to minimize skewness, but their distribution was still non-parametric. For baseline comparisons, generalized linear models were used, with linear, Poisson (for count data, e.g., MADRS scores), and gamma (for positively skewed distribution, e.g., serum CRP) distributions, as appropriate.

For the longitudinal analysis, due to the non-normal distribution of biomarkers and the clinical outcomes, generalized estimating equation (GEE) models were used. In analysis with biomarkers as an outcome, the best fit was found with gamma distribution with log link specification and an exchangeable covariance structure. For analyses with MADRS as the outcome, negative binomial models with log link specification and autoregressive covariance structure (AR-1) were selected. The independent variables were treatment group (i.e., infliximab vs. placebo), time (as a categorical variable), and the treatment × time interaction. The potential moderating effects of baseline self-reported childhood PA were also analyzed (e.g., treatment × time × childhood PA), in separate models. Due to the nonlinearity of the models, the estimated β coefficients were transformed into rate ratio (RR) estimates. No interim analysis was conducted.

## 3. Results

### 3.1. Demographics and Clinical Characteristics

A total of 55 participants were randomized at the MDPU site and were included in the analysis herein. Baseline sociodemographic and clinical characteristics of the intent-to-treat population are described in Table 1. There were no statistically significant demographic or clinical differences between groups. Twenty-four of 27 participants (88.8%) randomized to infliximab and 23/28 participants (85.7%) randomized to placebo received all three infusions. A total of 43 participants (78.2%) completed all 12 weeks; differences in study completion rates between treatment groups in this subsample were not statistically significant (*p* = 0.469). There were numerical, but not statistically significant (all *p*s > 0.1), differences in biomarkers (log transformed) at baseline between subjects in the infliximab and placebo groups.

### 3.2. Baseline Associations between NEV Biomarkers and Clinical Variables

There were no associations between age, sex, BMI, and use of tobacco and NEV biomarkers levels at baseline (all *p*s > 0.1). Use of psychotropic or antidiabetic medications was also not associated with any biomarker (all *p*s > 0.1). After adjustment for potential confounders, there were significant associations between biomarkers, MADRS and CRP, but no association between biomarkers and CTQ scores (Table 2).

### 3.3. Association of Infliximab with NEV Biomarker Changes

We initially assessed changes in NEV biomarkers over time in models adjusted only for Alix concentration, a canonical EV marker, to normalize for differential EV yield. There was no significant treatment by time interaction for any biomarker (all *p*s > 0.1). There was an effect of time only for TNFR2 (χ^2^ = 8.419, df = 3, *p* = 0.038), which showed an increase in levels in all subjects.

Subsequently, we conducted separate analyses to determine whether changes in NEV biomarkers were moderated by baseline PA as hypothesized a priori. There were moderating effects of PA on treatment by time interactions for TNFR1 (χ^2^ = 9.275, df = 3, *p* = 0.026), NF-κB (χ^2^ = 13.825, df = 3, *p* = 0.003), and IκBα (χ^2^ = 7.990, df = 3, *p* = 0.046). These models also indicated that there were significant treatment by time interactions for TNFR1 (χ^2^ = 10.305, df = 3, *p* = 0.016), NF-κB (χ^2^ = 9.095, df = 3, *p* = 0.028) and IκBα (χ^2^ = 7.900, df = 3, *p* = 0.048).

For TNFR1 levels, higher levels of PA were associated with larger decreases in infliximab-treated participants, relative to placebo, at week 2 (RR = 0.991, df = 1, *p* = 0.026), week 6 (RR = 0.987, df = 1, *p* = 0.009), and week 12 (RR = 0.983, df = 1, *p* = 0.003). Higher levels of PA were also associated with larger decreases in NF-κB levels in infliximab-treated participants, relative to placebo at week 6 (RR = 0.986, df = 1, *p* = 0.022) and week 12 (RR = 0.994, df = 1, *p* < 0.001), but not at week 2 (RR = 0.998, df = 1, *p* = 0.197). Similarly, larger decreases in IκBα levels in infliximab-treated participants, relative to placebo, were observed at week 12 (RR = 0.985, df = 1, *p* = 0.006), but not week 2 (RR = 0.995, df = 1, *p* = 0.2056) or week 6 (RR = 0.993, df = 1, *p* = 0.161).

Figure 1 illustrates the difference between placebo versus infliximab based on analysis of subgroups using the CTQ dichotomous clinical cut-off score that differentiates between the presence or absence of significant PA (i.e., ≥8) (16). There were no moderating effects of PA on treatment by time for c-Myc, FADD, IKKα/β, or TNFR2 (all *p*s > 0.1).

### 3.4. Moderation of Clinical Effect by NEV Biomarkers

Treatment effect on depressive symptom severity was previously reported. The biomarkers that were shown to change over time were then assessed as moderators of changes in depressive symptom severity. We observed a significant time by treatment by TNFR1 interaction on MADRS scores (χ^2^ = 7.997, df = 3, *p* = 0.046). In infliximab-treated patients, decreases in TNFR1 levels were associated with decreases in MADRS scores, more strongly at week 6 (RR = 0.672, df = 1, *p* = 0.005) than week 2 (RR = 0.849, df = 1, *p* = 0.985) or week 12 (RR = 0.801, df = 1, *p* = 0.128) (Figure 2). No other biomarker moderated changes in depressive symptoms.

### 3.5. Change in NEV Biomarkers and Cortical Thickness

In the subsample that underwent neuroimaging (*n* = 27), there were no baseline differences in age (*p* = 0.638), sex (*p* = 0.496), and MADRS total score (*p* = 0.720). Longitudinal analysis, adjusted for age and sex, indicated a trend for an increase in global cortical thickness in the infliximab group (treatment by time interaction: χ^2^ = 2.817, df = 1, *p* = 0.093). Changes in TNFR1 levels were associated with increased global cortical thickness in infliximab- (r = −0.581, *p* = 0.029) but not placebo-treated patients (r = 0.196, *p* = 0.501) (Figure 3). Changes in NF-κB or IκBα were not associated with change in global cortical thickness in both groups (all *p*s > 0.5).

## 4. Discussion

Herein, we leveraged circulating NEV biomarkers to show that infliximab engaged its target TNFR/NF-κB neuro-inflammatory pathway in individuals with BD, and that the molecular engagement in neurons was associated with clinical response and neurostructural changes. Interestingly, our results suggest that infliximab treatment resulted in inhibition of TNFR/NF-κB signaling in an early life trauma-dependent manner, similar to the overall effects in this clinical trial. Results from preclinical and clinical studies have consistently documented that TNF antagonists modulate NF-κB activity in diverse tissues [31,44,45,46,47,48]. TNF-α is a pro-inflammatory cytokine that exerts its effects by binding to two cell-surface receptors (TNFRs 1 and 2), which, through activation of downstream mediators, result in activation of NF-κB. Consequently, and consistently with our results, infliximab’s antagonism of TNF-α has been associated with downregulation of TNFRs, decreased activity of NF-κB, as well as of the mediators FADD, IKKα/β and IκBα. Although specific neuronal effects of TNF-α antagonists were previously shown in animal models [49,50], to the best of our knowledge, this is the first report of molecular effects in neurons of living humans using NEV isolation and analysis. We show that the NEV content of NF-κB signaling mediators downstream from TNFR are preferentially altered in the subset of infliximab-treated BD patients that also shows clinical response. This could be interpreted as a result of preferential engagement of the TNFα-TNFR-NF-κB signaling cascade in neurons of these patients and/or due to a preferential systemic response to infliximab, which also affects the neuronal response to it. Either way, neuronal signaling is the likely proximate cause of the clinical effect, and this is reflected as a differential change in NEV biomarkers.

Previous studies have documented an association between early life stress and increased NF-κB activation in PBMCs from individuals with major depressive disorder [51] and women with post-traumatic stress disorder [52], although in both studies NF-κB activity was more strongly correlated with symptom severity than with measures of childhood trauma. However, a separate study reported increased NF-κB activity following stimulation with lipopolysaccharides in PBMCs from adolescents exposed to childhood maltreatment, but without psychopathology, relative to controls with no history of trauma [53]. We did not observe an association between NEV biomarkers from the NF-κB pathways and CTQ scores at baseline. Nonetheless, our results were consistent with findings from the clinical trial, insofar as history of childhood maltreatment (i.e., physical abuse) predicted larger reductions in NF-κB pathway biomarkers in infliximab-treated participants, relative to placebo. These results support the hypothesis that neuroinflammation is a relevant pathophysiological mechanism connecting early life stress to BD, as well as a target for therapeutic interventions. Contrary to previously published work [54], we observed a negative association between NF-κB biomarkers and depressive symptoms severity at baseline. Of note, all participants had MADRS total scores of at least 22, indicating at least moderate depression, thus our study was not powered to reliably detect baseline associations between symptom severity and biomarker levels.

We also documented that changes in NEV TNFR1 levels moderated the antidepressant response to infliximab. In infliximab-treated participants, reductions in TNFR1 levels were associated with improvement of depressive symptoms, an effect that was not detected in the placebo group. Higher plasma levels of soluble TNFR1 have been consistently shown in BD [6,55,56,57,58,59]. Moreover, evidence indicates that increased plasma levels of soluble TNFR1 are associated with neurostructural abnormalities (e.g., reduction in hippocampal and gray matter volumes) in individuals with BD [6] and major depressive disorder [60]. Accumulating evidence indicates that TNFR-mediated TNF-α signaling is a key regulator of neuronal plasticity [61,62,63], particularly of synaptic scaling, which regulates the excitatory drive during chronic inactivity or hyperactivity [64,65]. In addition, evidence from preclinical studies indicates that activation of TNFR1 mainly leads to apoptosis and inflammation, whereas, in contrast, TNFR2 signaling mediates a homeostatic effect, including cell survival and regeneration [65,66]. While studies have documented that there is a complex interaction between TNFR1 and TNFR2, which seems to depend on many contextual factors (e.g., cell type, intracellular or extracellular environment), a reduction of TNFR1 activity within the context of preserved TNFR2 signaling could also shift the balance in favor of cellular survival [63,67,68,69,70,71]. Consistently with this hypothesis, we observed an association between reduction in TNFR1 levels in NEVs and increase in global cortical thickness in infliximab-treated participants. Thus, infliximab-induced effects on synaptic plasticity and neuronal excitability is one plausible mechanism to explain its antidepressant and neurostructural effects.

The current study also adds to the literature demonstrating the potential use of NEVs harvested from peripheral blood as a source of biomarkers to investigate target engagement and molecular responses to therapeutic interventions in clinical trials for neuropsychiatric disorders. There is evidence that neurons exposed to TNF-α release more EVs [72], suggesting that TNF-α regulates EV biogenesis and, potentially, that infliximab treatment could decrease the abundance of NEVs in plasma. However, in our study, we did not see an infliximab effect on NEV levels of Alix, an intra-vesicular EV marker, suggesting that, whatever effects infliximab might have had on neurons, it did not alter the levels of circulating NEVs. Moreover, the concentration of each NEV biomarker was normalized to that of Alix. Therefore, any biomarker differences observed cannot be attributed to an infliximab-induced change in NEV load between samples. Previously published work has used EVs to assess changes in biomarkers of brain insulin signaling in clinical trials of a protein restriction diet for prostate cancer [30], exenatide for PD [21], and intranasal insulin for AD [29]. This is the first NEV study focused on a neuro-inflammatory pathway in a mood disorders’ population, broadening the scope, and providing further support for the utility of NEVs as outcomes in clinical trials in neurology and psychiatry.

This study’s strengths include a participant population enriched for disturbed immune-inflammatory homeostasis, using biochemical and phenotypic criteria [15]. Although the immune-inflammatory criteria were heterogeneous, the resulting sample was more homogenous, and likely more powered to detect relevant effects of inflammatory processes, compared to the typical clinical samples of BD, composed usually of participants recruited solely based on current symptom severity. The convergent results vis-à-vis the modulatory effect of childhood maltreatment on a biological system, herein hypothesized a priori, informed by the results of this controlled clinical trial, reinforces the validity of these findings. Finally, this study’s longitudinal design and use of a therapeutic agent with a well-known biological target and a technique that allows for the direct measurement of molecular mediators in neuronal cells provides robust evidence regarding the role of neuroinflammatory processes in the pathophysiology, and potentially in the treatment, of BD.

Limitations include a relatively small sample size (*n* = 55), which makes the study unlikely to have been sufficiently powered to detect smaller effect sizes, and recruitment from a tertiary care clinic, which might limit its applicability to community samples. We also included participants receiving complex and mixed pharmacotherapy regimens; although we did not detect an association between specific agents and biomarkers levels, we could not completely rule out potential confounding effects. Moreover, our approach to NEV isolation has some limitations. We used immunoprecipitation targeting L1CAM, which has been widely accepted as a neuronal marker suitable for positive selection of NEVs due to its high expression by brain neurons (e.g., see recent publications by multiple groups [19,20,40,41]); however, it is also widely recognized that L1CAM is not exclusive to neurons (see https://www.proteinatlas.org/ENSG00000198910-L1CAM/tissue). Given that NF-κB signaling is also not specific to neurons, the potential contribution of non-neuronal but L1CAM+ EVs to the isolated EV subpopulation raises the possibility that the effects of infliximab on biomarkers cannot be solely attributable to neurons.

In conclusion, the results of this NEV biomarker study based on a randomized, placebo-controlled clinical trial indicates that infliximab engages its hypothesized neuronal target, in a trauma-dependent manner and in association with biologically and clinically meaningful results. This is the first study using NEVs to investigate the TNFR1/NF-κB pathway in neurons of living humans, as well as the potential neural effects of TNF-α antagonism, in individuals with BD. The evidence described herein can inform disease models of neuropsychiatric disorders centered on and/or involving neuroinflammatory mechanisms, in addition to providing further insights on the long-term clinical and biological effects of childhood maltreatment.

## Figures and Tables

**Figure 1 cells-09-00895-f001:**
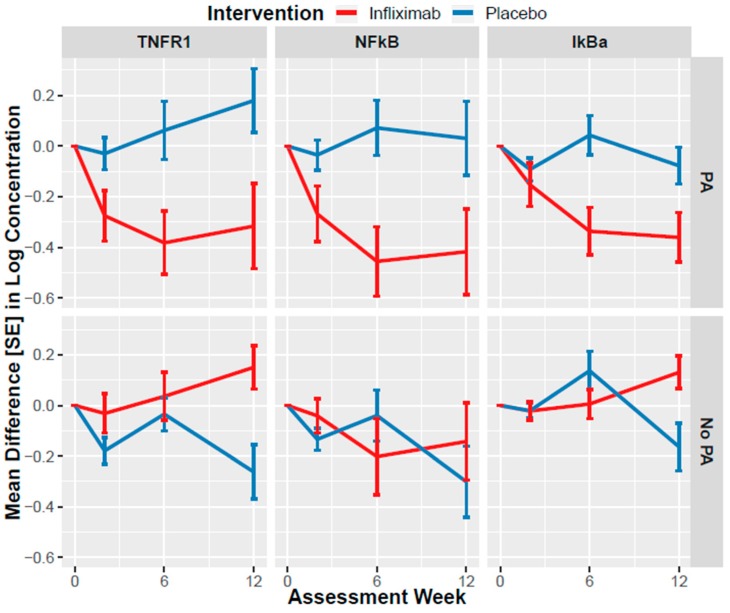
Least squares mean changes in biomarkers levels from baseline to week 12 in infliximab- or placebo-treated individuals with bipolar disorder with or without clinically significant history of physical abuse (PA). Results from an intent-to-treat generalized estimating equation analysis of 55 participants with bipolar disorder who were administered three infusions of infliximab (*n* = 27) or placebo (*n* = 28) at baseline and at weeks 2 and 6 of a 12-week trial. Error bars indicate standard errors (SE). Abbreviations: TNFR1: Tumour necrosis factor-alpha receptor-1; NF-κB: nuclear factor-kappa B; PA: with clinically significant history of physical abuse; No PA: without clinically significant history of physical abuse.

**Figure 2 cells-09-00895-f002:**
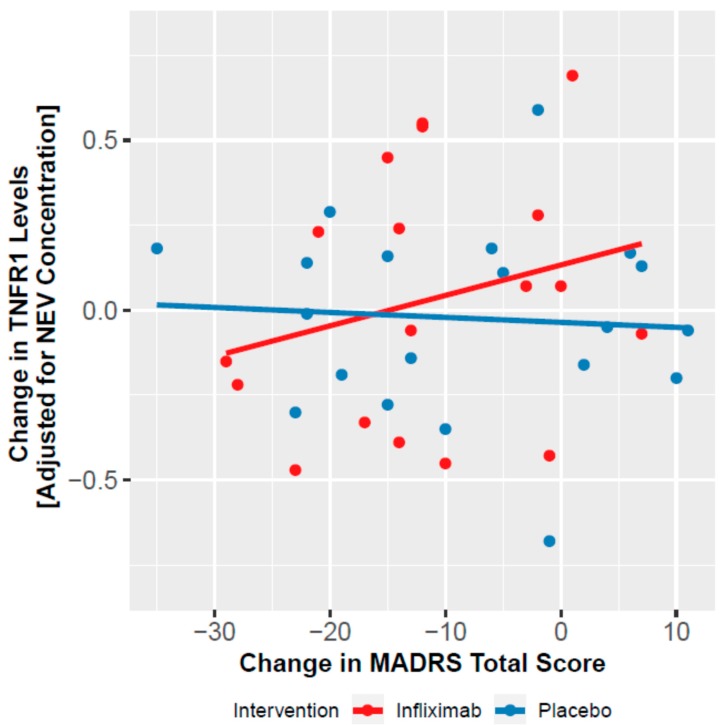
Association between changes in TNFR1 levels, adjusted for Alix concentration, and changes in MADRS scores, in the placebo and infliximab treated groups, at week 6.

**Figure 3 cells-09-00895-f003:**
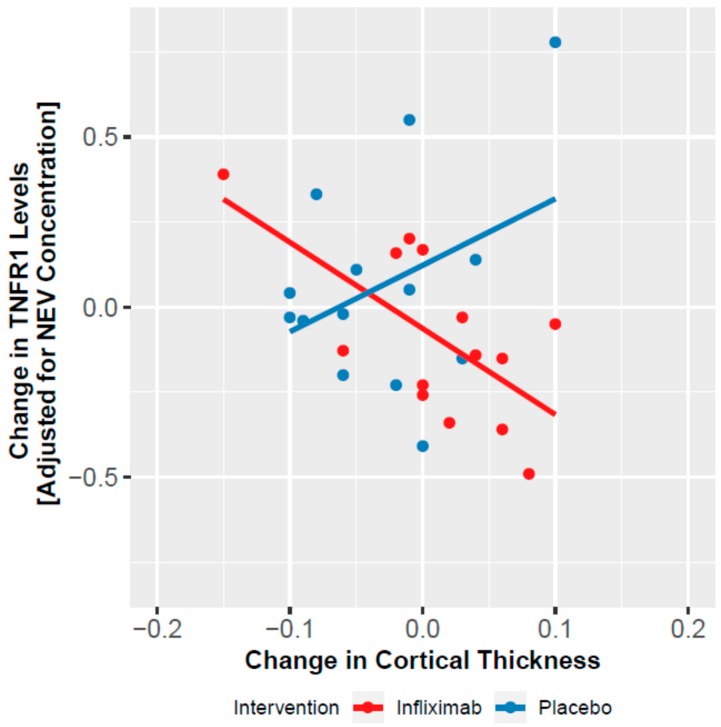
Association between changes in TNFR1 levels, adjusted for Alix concentration, and changes in cortical thickness, in the placebo and infliximab treated groups, at endpoint.

**Table 1 cells-09-00895-t001:** Sample baseline characteristics of the intent-to-treat population.

Baseline Characteristics	Placebo (*n* = 28)	Infliximab (*n* = 27)	*p*-Value
Age (years), mean (SD)	45.75 (10.28)	44.04 (11.55)	0.564 ^a^
Gender (female), *n* (%)	24 (85.7)	20 (74.1)	0.281 ^b^
Ethnicity (Caucasian), *n* (%)	5 (17.9)	4 (14.8)	0.760 ^b^
Education, *n* (%)			
High school	4 (14.8)	5 (19.2)	0.282 ^b^
College/University	22 (81.5)	17 (65.4)	
Graduate school	1 (3.7)	4 (15.4)	
MADRS (total score), mean (SD)	30.07 (6.72)	31.33 (6.85)	0.595 ^c^
YMRS (total score), mean (SD)	4.71 (4.30)	3.48 (3.08)	0.377 ^c^
BMI (kg/m^2^), mean (SD)	34.55 (7.66)	34.57 (10.08)	0.608 ^c^
Tobacco use, *n* (%)	8 (28.6)	10 (37.0)	0.504 ^b^
Age at onset (years), mean (SD)	17.14 (9.32)	18.59 (8.13)	0.437 ^c^
Number of lifetime psychiatric hospitalizations, mean (SD)	1.58 (2.02)	1.74 (1.91)	0.679 ^c^
Length of current depressive episode (months), mean (SD)	11.67 (20.80)	11.85 (15.48)	0.378 ^c^
Medications			
Antipsychotic, *n* (%)	16 (66.7)	18 (69.2)	0.846 ^b^
Antidepressant, *n* (%)	18 (75.0)	15 (57.7)	0.197 ^b^
Lithium, *n* (%)	5 (20.8)	6 (22.20	0.904 ^b^
Anticonvulsants, *n* (%)	10 (41.7)	15 (57.7)	0.258 ^b^
Antidiabetic, *n* (%)	4 (14.3)	4 (14.8)	0.956 ^b^
Childhood Trauma Questionnaire (total score), mean (SD)	54.46 (17.81)	56.51 (22.54)	0.866 ^c^
Physical Abuse, mean (SD)	9.35 (5.61)	8.48 (5.67)	0.266 ^c^

^a^*t*-test: ^b^ chi-square; ^c^ Mann-Whitney U.

**Table 2 cells-09-00895-t002:** Associations between NEV biomarkers and Montgomery–Asberg Depression Rating Scale (MADRS), C-reactive protein (CRP), and Childhood Trauma Questionnaire (CTQ), at baseline.

NEV	MADRS		CRP		CTQ Total Score	
	RR	*p*-Value ^a^	RR	*p*-Value ^b^	RR	*p*-Value ^c^
TNFR1	**0.944**	**0.014**	**1.176**	**0.026**	1.002	0.955
TNFR2	1.097	0.110	**2.021**	**<0.001**	1.190	0.072
NF-κB	**0.954**	**0.015**	**1.166**	**0.011**	1.047	0.156
c-Myc	0.961	0.100	**1.178**	**0.026**	1.042	0.288
FADD	0.962	0.098	**1.217**	**0.007**	1.067	0.094
IKKα/β	0.965	0.226	**1.385**	**<0.001**	1.048	0.346
IκBα	0.941	0.056	**1.389**	**<0.001**	1.080	0.136

^a^ Generalized linear model with Poisson distribution, adjusted for age, gender, and Alix concentration; ^b^ Generalized linear model with gamma distribution, adjusted for age, gender, and Alix concentration; ^c^ Generalized linear model with gamma distribution, adjusted for age, gender, MADRS total score, and Alix concentration.

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
