# Peer review of "Extracellular Vesicle Biomarkers Reveal Inhibition of Neuroinflammation by Infliximab in Association with Antidepressant Response in Adults with Bipolar Depression"

_cells, 2020, doi:10.3390/cells9040895_

Round 1

Reviewer 1 Report

The manuscript titled “Extracellular Vesicle Biomarkers Reveal Inhibition of Neuroinflammation by Infliximab in Association with Antidepressant Response in Adults with Bipolar Depression” in which the authors investigate the mechanistic mediators of infliximab’s effects, to improve depressive symptoms in a subpopulation of individuals with BD and history of childhood maltreatment. a very interesting and complex topic. the study is generally well designated, however the conclusions drawn by the authors have some limitations:

1-Did the authors consider or report differences in response to infliximab based on patient therapy? it should be clearly indicated or discussed in the text

2-the authors also should show the systemic effects of Infliximab on serum TNFa

3- section. Association of infliximab with NEV biomarkers changes. authors should show differences in responses in males and females. in two separate graphs beyond the cumulative graph shown

4-are the effects described by the authors dependent on a lower stimulated NEV tnf release? comment better in the discussion

Author Response

In section 3.2 we reported that no psychotropic agent affected the baseline levels of NEV biomarkers. Unfortunately, given the variety of treatments typically used in bipolar depression and our sample size we did not have enough power to reliably assess the effects of treatment on response and/or biomarkers.

In this study we have shown that the neuronal EV content of Nfkb signaling mediators downstream from TNFR are preferentially altered in the subset of infliximab-treated BD patients that also shows clinical response. This could be due to preferential engagement of the TNFa-TNFR- Nfkb cascade in neurons of these patients or due to a preferential systemic change of TNFa levels. The reviewer makes the really good suggestion that we should measure serum TNFa levels to help distinguish between these possibilities. Unfortunately, due to the SARS-coV-2 pandemic, our laboratory is closed indefinitely, and it is unlikely that we will be able to perform any additional experiments in any reasonable timeframe. We believe that this inability to follow the suggestion does not affect the validity of our findings, but only makes them amenable to a dual interpretation. Therefore, we now state in the discussion: “The nEV content of Nfkb signaling mediators downstream from TNFR are preferentially altered in the subset of infliximab-treated BD patients that also shows clinical response. This could be interpreted as a result of preferential engagement of the TNFa-TNFR- Nfkb signaling cascade in neurons of these patients and/or due to a preferential systemic response to infliximab, which also affects the neuronal response to it. Either way, neuronal signaling is the likely proximate cause of the clinical effect and this is reflected as a differential change in nEV biomarkers”.

Given our sample sex distribution, we did not have enough power to assess differences in response based on sex.

In response to this excellent point, we make the following addition to the discussion: “There is evidence that neurons exposed to TNF-a release more EVs (Russell et al., 2019), suggesting that TNF-a regulates EV biogenesis and, potentially, that infliximab treatment could decrease the abundance of NEVs in plasma. However, in our study, we did not see an infliximab effect on NEV levels of Alix, an intra-vesicular EV marker, suggesting that, whatever effects infliximab might have had on neurons, it did not alter the levels of circulating NEVs. Moreover, the concentration of each NEV biomarker was normalized to that of Alix. Therefore, any biomarker differences observed cannot be attributed to an infliximab-induced change in NEV load between samples.”

Reviewer 2 Report

In general this is an interesting and well-written manuscript. However, the inclusion of the neuroimaging are is not well justified and the authors provide no information on a putative mechanism by which they would expect neuroinflammatory signaling pathways to result in global changes in cortical thickness in the introduction.  This is particularly troubling given that the study included individuals over 60 years of age, where chronic mild inflammatory conditions are not uncommon and brain volumetric changes are well documented.  As presented, the neuroimaging arm seems largely divorced from the central focus of the manuscript, which is the moderation of anti-depressant actions of infliximab by childhood physical abuse.  In addition to the apparent lack relevance, several aspects of the image analysis are of concern:

  • Given the broad range in participant age, the authors failure to consider increased age as a factor in cortical gray matter loss is a distinct weakness of the analysis.
  • As the imaging arm of the study was accomplished in only a subset of participants, demographic information on these cohorts should be reported separately.
  • The differences noted in Figure 3 between infliximab and placebo conditions appear to be driven by two outliers. It appears that, were those removed, little difference would be seen between groups.
  • A much clearer and more comprehensive description of the extraction of the “global cortical volume” is needed, and a figure denoting the full extent of that volume should be provided

Author Response

All imaging analysis included age and sex as covariates, as described in section 3.5. This section also reports that there were no differences in demographic between the subsamples.

Formally, no values in Figure 3 can be classified as outliers, as they are all between -3 a +3 SD of their mean, so there is no rationale to exclude them.

Thank you for this comment, it should be noted that cortical thickness, not cortical volume, was used. We do however agree that a more comprehensive description of global cortical thickness is need. Unfortunately, as the variable is in reference to a 3D structure, it’s difficult to provide a figure that encapsulates the entirety of the measure. We hope the textual explanation is sufficient in describing the variable. The following has been added to the manuscript: “The global cortical thickness was obtained from the ‘Mean Thickness’ variable outputted by Freesurfer v6.0. This variable is a calculation, in millimeters, of the total thickness below the pial surface subtracted by the total thickness below the white surface, averaged across the entire surface of the brain. Therefore it measures the cortical thickness of only grey matter, found within the pial and white surface, within the left and right hemispheres (Fischl and Dale, 2000). Moreover, it excludes any subcortical volumes and the cerebellum.”

Reviewer 3 Report

Mansur et al., use circulating neuronally-derived extracellular vesicles as a marker to assess neuroinflammation in bipolar depression patients in a clinical trial of treatment with infliximab.  This is an important study on a novel subject manner. Assessing neuroinflammation using biomarkers has been difficult and this study utilizes circulating NEV cargo to “measure” neuroinflammation.  The NF-kB pathway and TNFRs were quantified from NEV at baseline and at 3 timepoints after treatment.  A reduction in TNFR1 levels in infliximab-treated patients was associated with an improvement in depressive symptoms.  Childhood physical abuse was associated with larger decreases in biomarker levels over time.  Neuroimaging was also performed on a subset of patients and TNFR1 levels were associated with increased global cortical thickness in infliximab treated patients. This study is informative and appropriate for publication in Cells.  

I have some comments that would strengthen the manuscript:

Was any characterization of the NEVs performed?  Were there any differences in NEV concentration? I understand there may be limited quantities of biomaterials.  If no characterization was performed, please state why in the methods.

The association between NEV biomarkers and CRP is strong at baseline.  Were there any differences over time?  Also, was there a decrease in CRP with treatment? 

Line 257 The CV >15 % between duplicate samples is really high.  Are these from independent NEV isolations?  Or the same NEV isolation run in duplicate? If it is the latter, this really doesn’t give much confidence to the data, especially if so many data points need to be excluded.  Please explain further.

Reference #15 is missing a reference. 

Symbols are weird in abstract and throughout the manuscript.

Figure 1 should state which one the blue and red lines are showing. 

Author Response

We have recently provided an extensive characterization of NEVs isolated from EDTA plasma using the same method employed in this study (published as supplemental material in Kapogiannis et al., 2019). From that publication, we are sharing NTA results (eFigure 1) showing that isolated NEVs (L1CAM+ EVs) have the typical size range of EVs, immunoblots (eFigure 3) showing the enrichment of transmembrane and intra-vesicular EV markers (Alix, CD81 and CD9) in total EVs and NEV preparations compared to EV-depleted plasma, as well as progressively diminishing levels of the Golgi-specific protein GM130, a negative EV marker, and of apoliptoprotein A1, a plasma lipoprotein component, from EV depleted plasma to Total EVs to L1CAM+ EVs. Unfortunately, due to the SARS-coV-2 pandemic, our laboratory is closed indefinitely, and it is unlikely that we will be able to perform
any additional experiments in any reasonable timeframe.

CRP decreased with treatment over time, but there was no association with changes in EV biomarkers.

We would like to clarify that CVs of NEV samples from the same isolation run in duplicate are < 15% for all biomarkers, which gives us confidence in the data. Specifically, we now clarify: “All assays were conducted in duplicate NEV samples from the same isolation and the respective coefficients of variation (CV) were 6.72% (TNFR2), 12.98% (NF-B), 12.65% (FADD), 14.33% (IKK/), 14.51% (IB), 13.50% (TNFR1), 12.54% (c-Myc), and 4.29% (Alix).” In addition to the above safeguard on data validity, to be even more conservative in our analysis, we identified and excluded the samples that had CVs of duplicates ≥ 15% separately for each assay. We have revised the statement for clarity: “Even though CVs of NEV samples from the same isolation run in duplicate were < 15% for all biomarkers, to be even more conservative in our analysis, , we identified and excluded the samples that had CVs of duplicates ≥ 15%, separately for each assay (17 samples for TNFR2, 35 samples for NF-B, 32 samples for FADD, 32 samples for IKK/, 37 samples for IB, 34 samples for TNFR1, 30 samples for c-Myc, and 5 samples for Alix). Therefore, reported results are likely a conservative underestimation of true effect sizes, given the loss of power from excluding these values.”

We fixed reference 15 and added a legend to Figure 1.

Round 2

Reviewer 1 Report

The authors responded adequately to all my points, considering the impossibility of carrying out new experiments at this time